# Effects of Human Adversarial and Affable Samples on BERT Generalization

**Aparna Elangovan[1], Jiayuan He[2,1], Yuan Li[1,2] and Karin Verspoor[2,1]**
[1]The University of Melbourne, Australia
[2]RMIT University, Australia
aparnae@student.unimelb.edu.au
{jiayuan.he, yuan.li, karin.verspoor}@rmit.edu.au

## Abstract

BERT-based models have had strong performance on leaderboards, yet have been demonstrably worse in real-world settings requiring generalization. Limited quantities of training data is considered a key impediment to achieving generalizability in machine learning. In this paper, we examine the impact of training *data quality*, not quantity, on a model's generalizability. We consider two characteristics of training data: the portion of human-adversarial (**h-adversarial**), i.e. sample pairs with seemingly minor differences but different ground-truth labels, and human-affable (**h-affable**) training samples, i.e. sample pairs with minor differences but the same ground-truth label. We find that for a fixed size of training samples, as a rule of thumb, having 10-30% h-adversarial instances improves the precision, and therefore $F_1$, by up to 20 points in the tasks of text classification and relation extraction. Increasing h-adversarials beyond this range can result in performance plateaus or even degradation. In contrast, h-affables may not contribute to a model's generalizability and may even degrade generalization performance.

## 1 Introduction

Deep learning models have dominated natural language processing (NLP) task leaderboards as seen in benchmarks such as GLUE (Wang et al., 2018), where the model performance is assessed using a held-out test set. However, the performance of these models may not be replicable outside of a given test set (Hendrycks et al., 2020), or they may not capture any relevant understanding of how language works (Bender et al., 2021). Benchmark test sets may not be sufficiently representative (Elangovan et al., 2021). These factors make it difficult to know when a model will work in practice and raise questions about model generalizability.

As an example of a possible disconnect between NLP models and what might be considered to be

| Training set | Alien language | Label |
|---|---|---|
| **Random** | timbuktu scio ghy | Positive |
| | shyp meyo mlue | Neutral |
| **H-adversarial** | timbuktu scio ghy | Positive |
| | timbuktu *meyo* ghy | Neutral |

Table 1: With **Random** training examples, a model or even humans can associate any one of the words "timbuktu", "ghy" or "scio" with positive label. Using the human adversarial pair (**H-adversarial**), it becomes more obvious that "scio" is an indicator for the positive label, and not "timbuktu" or "ghy".

language understanding, Pham et al. (2021) examined correct predictions of BERT-based classifiers, trained on GLUE tasks such as natural language inference (NLI). They found that 75% to 90% of these predictions were unchanged despite randomly shuffling words to render an input incoherent and blocking the inference. Elangovan et al. (2022) found that for a protein relation extraction model based on BioBERT (Lee et al., 2019), the precision dropped by over 50 points compared to test set performance when applied at scale to unseen data out of the official test set.

Conventional wisdom in machine learning is: the more data, the better. Limited training data is acknowledged as a key obstacle to achieving generalizability. In this paper, we study how *human adversarial samples* (**h-adversarials**) and *human affable samples* (**h-affables**) can impact generalisation performance. We define h-adversarials as samples that look very similar to each other, but have different ground-truth labels. We use the term "human adversarial" to differentiate it from adversarial machine learning which aims to fool the machines into making incorrect predictions by applying minor perturbations to a source sample, where the ground-truth label of the perturbed sample remains the same as its corresponding source (Goodfellow et al., 2015). We define h-affable samples as training samples that are very similar and have the same ground-truth label.

The intuition behind h-adversarials is to explicitly guide the model to learn discriminatory features through training data, thus reducing its reliance on spurious features (See Table 1). The concept of h-adversarials is also known by many other names such as counterfactual training data (Kaushik et al., 2020; Sen et al., 2021) or contrast sets (Gardner et al., 2020). We also hypothesize, that h-affable training samples, or samples that are similar to another but have minor differences that preserve the label, can potentially result in redundant information in the training data, reinforcing possibly spurious patterns and worsening a model's performance. Therefore, annotating large volumes of training data with a high ratio of h-affables can potentially make the data annotation efforts ineffective.

Data augmentation strategies for adding training samples by making minor perturbations that preserve (h-affables) or alter (h-adversarials) instance labels have been shown to improve model robustness (Wang et al., 2022). Here, we explore the question, "*How do various proportions of h-affables and h-adversarials in training samples affect the model's generalization performance?*" We examine three NLP classification tasks, including text classification, relation extraction, and keyword detection, and perform an empirical study on the impact of h-adversarial and h-affable samples. The experimental results confirm our hypothesis – with fixed training data size, a model's performance can exhibit a significant improvement with a moderate proportion of h-adversarial samples, while h-affable samples can contribute to lowering model performance. These results reveal how data *quality*, not data *quantity* alone, may affect model performance, thereby providing insights into data annotation and curation procedures. With limited data annotation budgets, creating high-quality training samples is critical to guide a model towards learning discriminative features. Ultimately, this will lead to robust generalization performance.

## 2 H-adversarials and h-affables

Given a source sample $s$ assigned with the ground-truth label $l$, we define a h-adversarial sample $\tilde{s}_{l'}$ as a sample generated by applying $\delta$ changes to $s$ such that $l$ is changed to a different label $l'$, where $\delta$ is less than a pre-defined threshold $\epsilon$:

$$s_l \xrightarrow{\delta < \epsilon} \tilde{s}_{l'} \mid l \neq l' \qquad (1)$$

Similarly, we define a h-affable sample $\tilde{s}_l$ as a sample generated by applying $\delta$ changes to $s$ but the ground-truth label $l$ remains the same:

$$s_l \xrightarrow{\delta < \epsilon} \tilde{s}_l \qquad (2)$$

In this paper, we focus on the tasks of text classification and relation extraction, where a sample corresponds to a text. Given a reference sample $s$ and another sample $\tilde{s}$, we measure their input difference $\delta^{s,\tilde{s}}$ using the word error rate based on Levenshtein edit distance (Miller et al., 2009):

$$\delta^{s,\tilde{s}} = \frac{I + D + S}{|w_s|}$$

where $I$ is the number of insertions, $D$ is the total number of deletions, $S$ is the total number of substitutions required to transform sample $s$ to sample $\tilde{s}$, and $|w_s|$ is the total number of words in the reference sample $s$. Note that other distance metrics can also be adopted here. However, we keep the distance metric simple and explainable to examine the learned features of deep learning models.

Given a training set for an NLP classification task, we define the h-adversarial rate $r_{\text{hv}}^{ll'}$ for the class label $l$ and $l'$ as the portion of samples in class $l'$ that have an h-adversarial sample in class $l$:

$$\tilde{n}_{l'} = \Sigma_{i'=1}^{|S_{l'}|} \begin{cases} 1, & \text{if } \exists i \in [1, |S_l|] : \delta^{s^i, s^{i'}} < \epsilon \\ 0 & \text{otherwise} \end{cases}$$

$$r_{\text{hv}}^{ll'} = \frac{\tilde{n}_{l'}}{|S_l|} \qquad (3)$$

Here, $S_l$ is the set of training samples labelled with $l$, $S_{l'}$ is the set of samples labelled with $l'$, and $\tilde{n}_{l'}$ is the number of samples labelled with $l'$ that are within $\epsilon$ edit distance to at least one sample labelled with $l$, i.e., can be derived by applying less than $\epsilon$ changes to another sample with label $l$.

The value of $r_{\text{hv}}^{ll'}$ can range between $[0, |S_{l'}|]$, where 0 indicates that there are no h-adversarial samples for label $l \to l'$. When $r_{\text{hv}}^{ll'}$ is greater than 1, it indicates that a single source sample $s_l$ can have multiple h-adversarial samples $\tilde{s}_{l'}$. It's worthwhile to note that in such cases, these h-adversarials are all similar to $s_l$ in terms of input, and may h-affables for each other.

We define h-affable rate as $r_{\text{hf}}^l$ for label $l$ as follows:

$$\tilde{n}_l = \Sigma_{i=1}^{|S_l|} \begin{cases} 1, & \text{if } \exists j \in [i+1, |S_l|] : \delta^{s^i, s^j} < \epsilon \\ 0 & \text{otherwise} \end{cases}$$

$$r_{\text{hf}}^l = \frac{\tilde{n}_l}{|S_l|} \qquad (4)$$

where $\tilde{n}_l$ is the number of samples that can be transformed by applying less than $\epsilon$ changes from another sample with the same label. The value of $r_{\mathrm{hf}}^l$ can range between $[0, 1]$.

The h-adversarial and h-affable samples can be curated by domain experts, and also can be generated automatically in some cases if appropriate rules can be defined given the context of a task. Taking a binary relation extraction task as an example, a given source text annotated with participating entities can be used to generate multiple samples by marking different entities as involved in the relationship, as shown in Table 2. For the given text with 11 words with 4 entities mentioned, $C_4^2 = 6$ training samples can be automatically generated. At most 4 substitutions are required to change one sample to another, with maximum sample difference of $\delta^{s,\tilde{s}} = \frac{4}{11} = 0.36$.

## 3    Method

We examine the impact of h-adversarials and h-affables on three different NLP classification tasks, including text classification, relation extraction, and a customized keyword detection task (detailed in Section 3.1.2). Next, we introduce these tasks and datasets, focusing on the generation of h-adversarials and h-affables in these tasks.

### 3.1    Tasks and datasets

We describe three main tasks, evaluated in several datasets. These are summarized in Table 3.

### 3.1.1    Counterfactual IMDB

This task aims to predict the binary sentiment (positive or negative) of a given user review. We use the IMDB dataset (IMDB-C) created by Kaushik et al. (2020). The original dataset (Kaushik et al., 2020) has around 3400 samples. From the original dataset we randomly sample 2000 samples (IMDB-C-2K) and 500 samples (IMDB-C-5H) for each run, and this is repeated for 5 different runs. To evaluate the generalizability of a model, we use three datasets that provide labelled user reviews in different domains other than IMDB: (a) Amazon polarity test set[1]; (b) Yelp polarity test set[2]; (c) SemEval-2017 twitter dataset for Task4-B test set[3] (Rosenthal et al., 2017) (d) IMDB contrast test set

---

[1]https://huggingface.co/datasets/amazon_polarity
[2]https://huggingface.co/datasets/yelp_polarity/tree/main
[3]https://alt.qcri.org/semeval2017/task4/?id=download-the-full-training-data-for-semeval-2017-task-4

(Cont)[4] (Gardner et al., 2020). To test the impact of h-adversarial samples, we use the manually curated h-adversarial samples that come with the official training set.

### 3.1.2    Self-Labelled Keywords And-Or detection

The Keywords And-Or detection (KDAO) task is a binary text classification (CLS) task that we define in this paper, where labels can be generated automatically. As such, we can generate training and test samples automatically, as well as the h-affables and h-adversarials. This allows us to evaluate model performances on data at scale.

For this task, we assume two non-overlapping sets of keywords, $K_a$ and $K_b$. Given an input text $W$ with $n$ words $\langle w_1, w_2, ..w_i, w_n \rangle$, a positive label is assigned, if and only if the input text (1) contains at least two words from set $K_a$ **and** (2) at least 1 word from set $K_b$. Hence, this mimics a combination of Boolean *And* and *Or* operators resulting in the equation shown below. The pseudo-code is available in Appendix 1.

$$match_a(W) = \vee_{i,j=1, w_i \neq w_j}^n w_i \in K_a \wedge w_j \in K_a$$
$$match_b(W) = \vee_{i=1}^n w_i \in K_b$$
$$label(W) = match_a(W) \wedge match_b(W)$$

Although KDAO is a rule-based task that is seemingly naive, surprisingly, our experiments show that this task is a non-trivial task for BERT, requiring over 8,000 training samples to reach 80% F1 (see Figure 4) compared to IMDB sentiment analysis, which only requires 2000 samples to reach over 80% accuracy as shown in Table 4.

We create the dataset for this task using the collection of PubMed abstracts[5] that contains 18 million abstracts of medical journals. For training, we filter and only keep abstracts with 100 to 250 words, from which we sample 2400 abstracts, resulting in a training set with a positive sample rate of 25%. We use 2,000 samples for training and set aside 400 abstracts as the validation set. To test the generalizability of a model, we create a very large test set by randomly sampling 500,000 PubMed abstracts from the collection.

**Automatic generation of h-adversarials & h-affables.** The h-adversarial/h-affable samples in the KDAO task can be generated automatically.

---

[4]https://github.com/allenai/contrast-sets/blob/main/IMDb/data/test_contrast.tsv
[5]https://ftp.ncbi.nlm.nih.gov/pubmed/baseline/

| # | Type | label | Text |
|---|------|-------|------|
| - | O | - | Gene **NLCR** *inhibits* **KLK3** and **EFGR**, but has no effect on **MAPK**. |
| REL-1 | $s_p$ | P | Gene **MARKER-A** *inhibits* **MARKER-B** and **EFGR**, but has no effect on **MAPK**. |
| REL-2 | $\tilde{s_n}$ | N | Gene **MARKER-A** *inhibits* **KLK3** and **EFGR**, but has no effect on **MARKER-B**. |
| REL-3 | $\tilde{s_n}$ | N | Gene **NLCR** *inhibits* **MARKER-A** and **EFGR**, but has no effect on **MARKER-B**. |
| REL-4 | $\tilde{s_p}$ | P | Gene **MARKER-A** *inhibits* **KLK3** and **MARKER-B** , but has no effect on **MAPK**. |
| - | O | - | The lasagne was awesome, so was the spagetti. |
| CLS-1 | $s_p$ | P | The lasagne was **awesome**, so was the spagetti. |
| CLS-2 | $\tilde{s_n}$ | N | The lasagne was **under cooked**, so was the spagetti. |

Table 2: Examples of how a single **O**riginal sample can be transformed to **P**ositive and **N**egative samples either automatically (e.g., binary **rel**ation extraction) or with domain expertise (e.g., text **cla**ssification). $s_p$: original sample with positive label; $\tilde{s_p}/\tilde{s_n}$: transformed samples with minor changes to $s_p$. For relation extraction, we follow previous works (Zhang et al., 2019) to mark participating entities with special markers (MARKER-A and MARKER-B). For relation extraction, REL-1 ($s_p$) has 2 h-adversarials, REL-2 ($\tilde{s_n}$) and REL-3 ($\tilde{s_n}$). REL-2 and REL-3 also act as h-adversarials for REL-4 ($\tilde{s_p}$). REL-2 and REL-3 are also a pair of h-affables for the label N, while REL-1 and REL-4 are a pair of h-affables for label P. For text classification (sentiment classification), CLS-2 forms an adversarial sample for CLS-1 if threshold $\epsilon$ is set to 0.25.

| Dataset | Task | Tr+% | Train | Te+% | Test |
|---------|------|------|-------|------|------|
| ChemProtC3 | REL | 27.0 | 767 | 11.6 | 5744 |
| KDAO | CLS | 25.0 | 2000 | 4.8 | 500000 |
| IMDB-C-5H-Amzn | CLS | 50.0 | 500 | 50.0 | 400000 |
| IMDB-C-5H-Yelp | CLS | 50.0 | 500 | 50.0 | 38000 |
| IMDB-C-5H-Sem | CLS | 50.0 | 500 | 50.0 | 6000 |
| IMDB-C-5H-Cont | CLS | 50.0 | 500 | 50.0 | 488 |
| IMDB-C-2K-Amzn | CLS | 50.0 | 2000 | 50.0 | 400000 |
| IMDB-C-2K-Yelp | CLS | 50.0 | 2000 | 50.0 | 38000 |
| IMDB-C-2K-Sem | CLS | 50.0 | 2000 | 50.0 | 6000 |
| IMDB-C-2K-Cont | CLS | 50.0 | 2000 | 50.0 | 488 |

Table 3: All datasets have two classes: positive and negative. The percentage of samples with positive labels is reported for training (Tr+%) and test (Te+%) data.

Given a source sample with a positive label, an h-adversarial sample can be generated by replacing the word that matches with $K_a/K_b$ with a random non-keyword word so the label changes from Positive to Negative. The sample code is available in Appendix A.2. To generate h-affable samples, we insert a random word into the abstract at a random location. Since the abstract will still contain the key words to be detected, the label remains to be Positive. The full code is available in Appendix A.3.

### 3.1.3 ChemProt

The ChemProt dataset (Krallinger et al., 2017) is curated for a relation extraction task (REL) for extracting 6 types (5 types of positive + 1 negative) of relationship between chemicals and proteins from PubMed abstracts. In order to simplify the task into a binary classification problem, we select one class (annotated as "CPR:3" in the original dataset) as the positive class and re-annotate the dataset by mapping the remaining classes as negative examples. We refer to this dataset as ChemProt-C3.

**Automatic generation of h-adversarials & h-affables.** We transform positive samples to h-adversarial negatives following the method presented in Table 2: shuffling the positions of the markers that are used to mark the participating entities of the relationship. The affable samples are samples from the same abstract with the same ground-truth label but have different entities participating in the relationship as shown in Table 2.

### 3.2 Models and experiment settings

We employ BERT models for all tasks, selecting the appropriate variant based on the corpus domain of each dataset. For IMDB datasets, we use the standard BERT-base-cased model (Devlin et al., 2019) as well as RoBERTa-base (Liu et al., 2020). For KDAO and ChemProt-C3, which are from the biomedical domain, we employ BioBERT (Lee et al., 2019), a variant of BERT model that is further trained on PubMed abstracts using the masked language modelling task.

In our experiments using BERT and BioBERT, we use a learning rate of 1e-05. For RoBERTa, we use a learning rate of 7e-06. We use a batch size of 64 with gradient accumulation, with early stopping patience of 10 epochs with maximum epoch of 200 iterations. We set the distance threshold for computing h-adversarial and h-affable rate as 0.25. In order to account for BERT stability (Mosbach et al., 2021), for each set of experiments, we repeat our training procedure with different random seeds 5 times and report the results of all runs.

## 4 Experiments and results

**Experiment-A: Varying h-adversarials rate.** In this set of experiments, we study the effect of h-

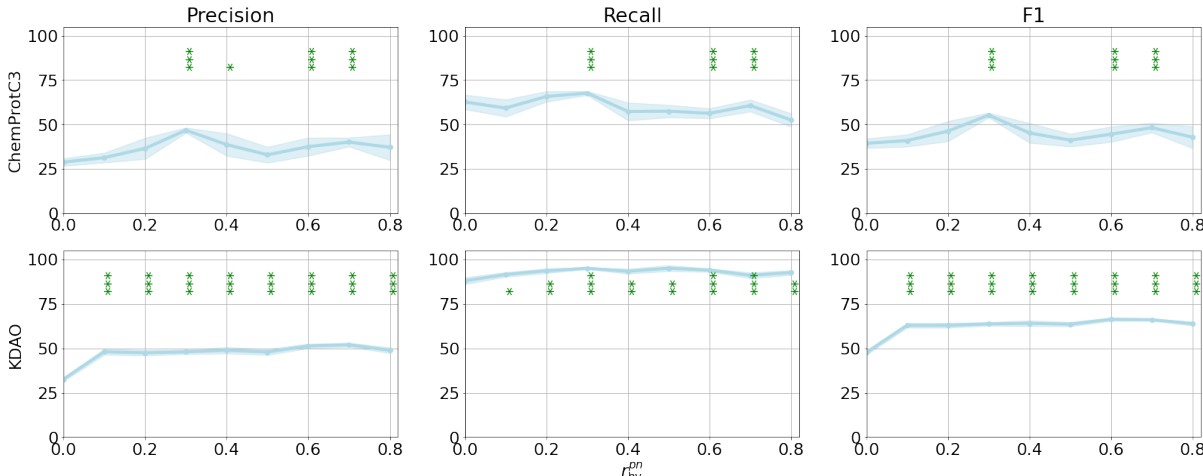

Figure 1: Impact of h-adversarial rate for Positive class ($r_{\mathrm{hv}}^{pn}$) on fixed-size training data on BioBERT. Precision starts to increase by 10 to 20 points around 10% to 30% h-adversarial rate. Shaded region shows standard error. Indicators * p-value < 0.1, ** p < 0.05, *** p < 0.001 significance of performance compared to the case where $r_{\mathrm{hv}}^{pn} = 0.0$.

adversarial rate, where we hold the training size and the positive sample rate constant. To introduce h-adversarial samples to the training set, we randomly choose a subset of positive samples from existing training samples, transforming them to negative samples to obtain corresponding h-adversarial samples (see Section 3.1.3). To maintain the size of training data and the positive sample rate, we then randomly drop existing negative samples. The h-affables rate is held constant at zero in this set of experiments.

**Result-A: For a fixed training size, low to moderate h-adversarial rates improve performance**: Table 4 on BERT shows that for the IMDB-C-2K and IMDB-C-5H datasets, the peak accuracy is achieved at around $@r_{hv}^{pn} = 0.1$, where we see marginal improvements (up to 2 absolute points) for Amazon and Yelp test sets, but a 7-point improvement for SemEval test. However, for the IMBD-contrast test set, we find that the peak performance is at $@r_{hv}^{pn} = 0.9$, as shown in Table 4. For RoBERTa, as shown in Table 5, $@r_{hv}^{pn} <= 0.3$ generally has the highest performance for all datasets, except SemEval where the peak performance is at $@r_{hv}^{pn} = 0.0$. KDAO dataset also achieves a peak performance at $@r_{hv}^{pn} = 0.1$, increasing precision from 32.7 ($\sigma_M$ 1.1) $\rightarrow$ 48.1 ($\sigma_M$ 1.7), thus increasing F1 from 47.6 ($\sigma_M$ 1.3) $\rightarrow$ 62.9 ($\sigma_M$ 1.3), where $\sigma_M$ represents the standard error. For the ChemProt-C3 dataset, the precision increases steadily from $@r_{hv}^{pn} = 0.0 \rightarrow @r_{hv}^{pn} = 0.3$, achieving its peak precision of 46.8 ($\sigma_M$ 1.4) thus in-

| Test | $r_{\mathrm{hv}}^{pn}$ | IMDB-C-5H | IMDB-C-2K |
|---|---|---|---|
| | 0.0 | 84.0 (0.8) | 88.2 (0.6) |
| | 0.1 | **86.2 (0.4)** | **88.9 (0.4)** |
| Amazon | 0.2 | 85.2 (0.9) | 88.4 (0.3) |
| | 0.3 | 85.2 (1.2) | 88.7 (0.7) |
| | 0.9 | 85.1 (1.3) | 86.9 (0.7) |
| | 0.0 | 79.9 (2.0) | 86.5 (1.0) |
| | 0.1 | 86.3 (0.4) | 89.2 (0.3) |
| Cont | 0.2 | 86.1 (0.9) | 89.2 (0.2) |
| | 0.3 | 87.0 (1.3) | 89.3 (0.3) |
| | 0.9 | **91.3 (0.5)** | **92.9 (0.3)** |
| | 0.0 | 70.2 (5.4) | 73.0 (3.6) |
| | 0.1 | **77.0 (1.7)** | **77.8 (1.4)** |
| SemEval | 0.2 | 72.4 (2.1) | 70.3 (3.1) |
| | 0.3 | 75.0 (3.8) | 73.7 (3.8) |
| | 0.9 | 67.9 (4.6) | 70.4 (1.1) |
| | 0.0 | 85.3 (0.6) | 90.0 (0.4) |
| | 0.1 | **87.5 (0.5)** | **90.1 (0.2)** |
| Yelp | 0.2 | 86.0 (1.1) | 89.9 (0.3) |
| | 0.3 | 85.0 (2.8) | 90.0 (0.5) |
| | 0.9 | 85.5 (1.6) | 87.0 (0.7) |

Table 4: Performance in Accuracy ($\sigma_M$) on BERT when varying $r_{\mathrm{hv}}^{pn}$ in IMDB-C fixed size training set and evaluated on Amazon, SemEval, IMDB-Contrast test set and Yelp. (Note: $\sigma_M = \frac{\sigma}{\sqrt{n}}$ = Standard error)

creasing F1 from 39.4 ($\sigma_M$ 2.7) $\rightarrow$ 55.3 ($\sigma_M$ 1.3) as shown in Figure 1. Increasing $r_{hv}^{pn}$ beyond this does not improve F1 or accuracy for any of the datasets as shown on Figure 1 and Table 4, and in fact IMDB-C and and ChemProt-C3 seem to drop performance. Increasing h-adversarial rates does

| Test | $r_{\text{hv}}^{pn}$ | IMDB-C-5H | IMDB-C-2K |
|---|---|---|---|
| Amazon | 0.0 | 76.0 (4.2) | **90.6 (0.4)** |
|  | 0.1 | 68.1 (6.3) | 90.0 (1.1) |
|  | 0.2 | 75.6 (6.8) | 86.5 (2.0) |
|  | 0.3 | **85.9 (1.1)** | **90.6 (0.7)** |
|  | 0.9 | 78.0 (7.2) | 86.4 (2.5) |
| Cont | 0.0 | 73.9 (7.3) | 89.4 (0.4) |
|  | 0.1 | 66.6 (9.1) | 92.2 (0.4) |
|  | 0.2 | 82.1 (8.0) | 90.0 (2.4) |
|  | 0.3 | **92.2 (0.6)** | **92.0 (0.4)** |
|  | 0.9 | 85.0 (8.9) | 91.8 (2.5) |
| SemEval | 0.0 | **62.7 (1.5)** | **76.6 (1.9)** |
|  | 0.1 | 51.6 (4.6) | 68.7 (4.4) |
|  | 0.2 | 60.3 (5.0) | 63.2 (3.3) |
|  | 0.3 | 57.7 (1.6) | 65.6 (4.6) |
|  | 0.9 | 58.0 (5.9) | 63.7 (4.6) |
| Yelp | 0.0 | 78.4 (4.6) | 91.9 (0.3) |
|  | 0.1 | 68.9 (6.6) | 91.3 (0.8) |
|  | 0.2 | 77.7 (7.1) | 88.3 (2.0) |
|  | 0.3 | **86.7 (0.9)** | **92.2 (0.5)** |
|  | 0.9 | 78.5 (7.5) | 87.0 (2.4) |

Table 5: Performance in Accuracy ($\sigma_M$ ) on RoBERTa when varying $r_{\text{hv}}^{pn}$ in IMDB-C fixed size training set and evaluated on Amazon, SemEval, IMDB-Contrast test set and Yelp.

not seem to significantly impact recall as shown in Figure 1 ChemProt-C3 and KDAO datasets.

**Experiment-B: Varying h-affables rate.** In this set of experiments, we study the effect of h-affable rate. We also hold the training size and the positive sample rate constant in this setting, and only increase the h-affables rate. We further consider 2 cases: varying the h-affable rate for Positive and Negative class. To vary the h-affables rate for positives, we generate affable positive samples using the method detailed in Section 3.1 and randomly drop non-affable positives to maintain the positive sample rate, and vice versa for varying h-affable samples for negatives. In both scenarios, the h-adversarial rate is held constant at zero.

**Result-B: For a fixed training size, high h-affables rates for Positive class eventually degrade recall while high h-affable rates for Negative class eventually degrade precision.** Figure 2 and Figure 3 show that the h-affable rate for Positive class affects the recall, while the h-affable rate for Negative class affects the precision of a model. We see that when the h-affable rate for positives increases beyond 0.3, recall starts to drop in both tasks. For the negative sample h-affable rate, the

precision drops at around 0.6. We also find that at lower range of h-affable rates, e.g., between 0.0 and 0.3, the difference in h-affable rate does not seem to make significant difference to the performance.

**Experiment-C: Varying training size.** In this set of experiments, we study the learning curves of models over experience (amount of training data), by gradually adding more samples to the training set. We consider three ways of adding samples: (a) adding more training data that are random samples where the h-adversarial and h-affable rates are held at zero; (b) only adding positive and negative h-affable samples where the h-adversarial rate is kept as zero; and (c) adding a mixed set of h-adversarial samples and random samples where the h-adversarial rate @$r_{\text{hv}}^{pn}$=0.1 and the h-affable rate is kept as zero (@$r_{\text{hf}}^{p}$=0.0, @$r_{\text{hf}}^{n}$=0.0). For all scenarios, we keep the positive sample rate constant when adding more samples.

**Result-C: When baseline (@$r_{\text{hv}}^{pn}$=0.0) performance is low, adding smaller percentages of h-adversarials boosts performance compared to adding random samples while adding h-affables make almost no difference.** When baseline dataset (@$r_{\text{hv}}^{pn}$=0.0) performance is relatively low (in our experiments less than 80% score), we see that adding 10% h-adversarials can substantially improve performance. This is observed for the KDAO dataset as shown in Figure 4, where the F1 improves by up to 20 points compared to adding random samples. As the F1 reaches approximately 80% or above, the performance improvements through adding h-adversarials are similar to adding random samples in Figure 4. For the IMDB-C-5H evaluated on SemEval @$r_{\text{hv}}^{pn}$=0.0, we see a strong boost 70.2 ($\sigma_M$ 5.4) $\rightarrow$ 77.0 ($\sigma_M$ 1.7), while the boost for Amazon and Yelp is relative small, 84.0 ($\sigma_M$ 0.8) $\rightarrow$ 86.2 ($\sigma_M$ 0.4) and 85.3 ($\sigma_M$ 0.6) $\rightarrow$ 87.5 ($\sigma_M$ 0.5) respectively as shown in Table 4. The improvements in performance compared to baseline for Amazon, SemEval and Yelp further narrow down for IMDB-C-2K compared to IMDB-C-5H as shown in Table 4. From Figure 4, we observe that adding more h-affables to training data does not seem to improve the model, even when the training size increases from 2,000 to 10,000 samples on the KDAO task. This demonstrates that adding more similar training data that preserve the label need not improve the performance.

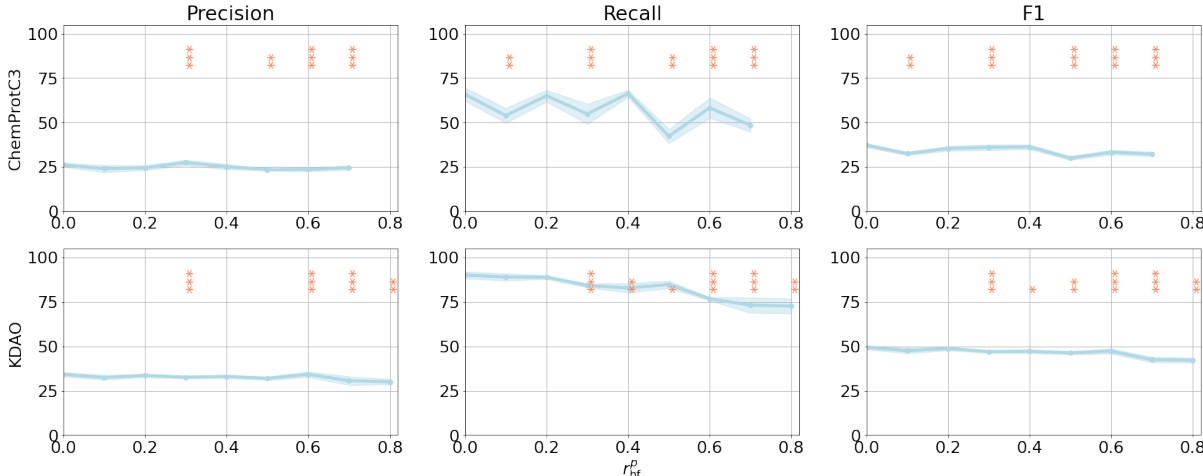

Figure 2: With fixed size training data, high h-affable rate for Positive class **drops recall**. Recall starts to drops around h-affable rate of 0.3. Shaded region shows standard error. Indicators * p-value < 0.1, ** p < 0.05, *** p < 0.001 compared to $r_{\text{hf}}^{p} = 0.0$ performance. **(Bottom)** Performance on KDAO task. **(Top)** Performance on ChemProt-C3.

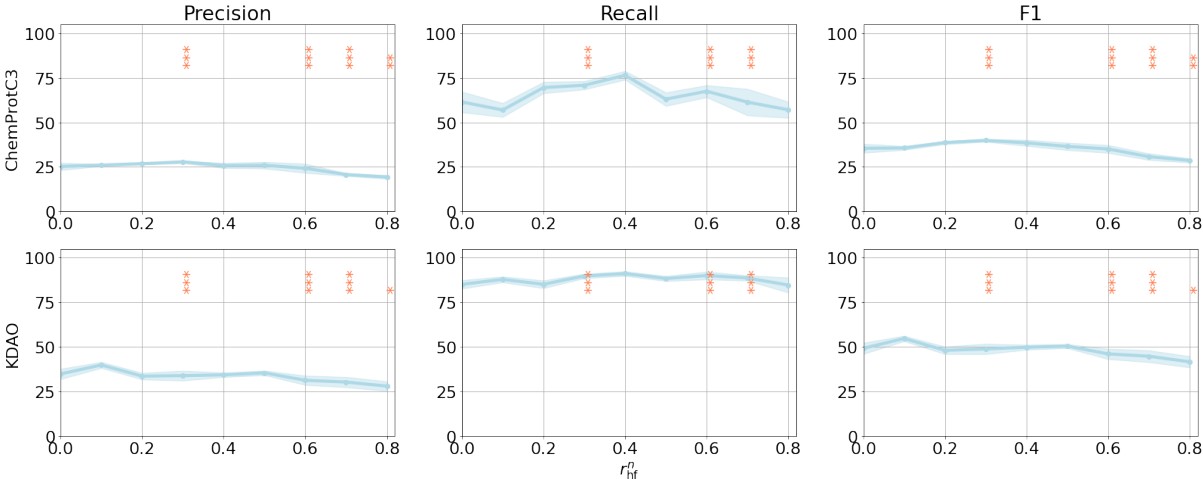

Figure 3: With fixed size training data, high h-affable rate for Negative class **drops precision**. Precision drops around h-affable rate of 0.6. Shaded region standard error. Indicators * p-value < 0.1, ** p < 0.05, *** p < 0.001 significance compared to $r_{\text{hf}}^{n} = 0.0$ performance. **(Bottom)** Performance on KDAO task. **(Top)** Performance on ChemProt-C3.

## 5 Discussion

### 5.1 Directing data curation efforts

Our experiments show that smaller percentage of h-adversarials have the potential to substantially boost performance compared to adding random samples. Increased proportions of h-adversarials beyond about 30% do not result in performance improvements or can even degrade performance compared to adding random samples. Given that manually creating h-adversarial samples is more time consuming than simply annotating random samples, it may be most effective to add a small percentage of h-adversarials during data curation.

Augmenting training data with h-affables leads to almost no improvement even when increasing the volume from 2,000 to 10,000 samples. When h-affables have to be manually annotated, this would imply wasted resources. The lack of improvement from adding h-affables also shows that data augmentation strategies that rely on generating more similar samples may not improve generalizability. Our results also support findings that data augmentation strategies using synonym replacement or random swaps/insertions/deletions may not improve performance much beyond 2000 samples when evaluated on a text classification task (Wei and Zou, 2019). For a fixed amount of training data, a higher

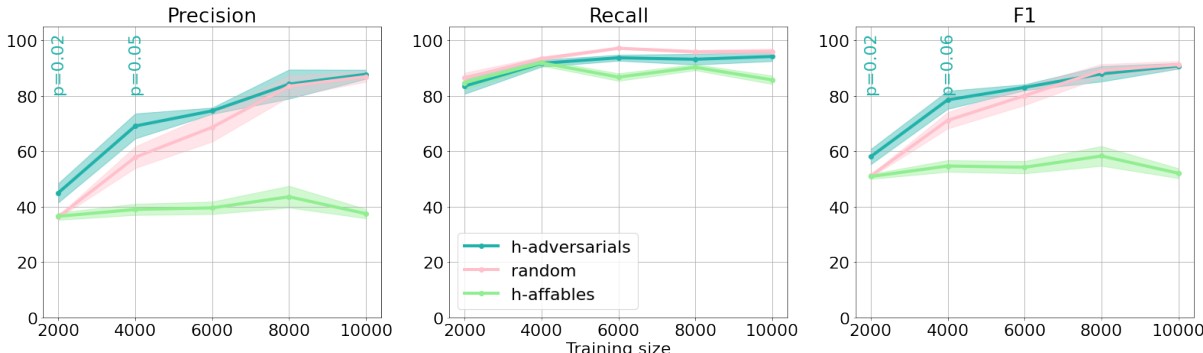

Figure 4: Impact of adding training data evaluated on KDAO task with 500,000 test samples. Adding h-adversarials ($r_{hv}^{pn} = 0.1$) with random samples improves performance more quickly when the number of training samples is limited (<6,000). Adding diverse random samples ($r_{hv}^{pn} = 0$, $r_{hf}^{n} = 0$, $r_{hf}^{p} = 0$) increases performance almost linearly for precision and therefore $F_1$. Recall sees marginal improvements. Adding only positive and negative h-affables make very little or almost no difference to improvements despite increasing the training size from 2,000 to 10,000. Shaded region shows standard error, p-value compares random to h-adversarial performance.

h-affable rate for positive samples degrades recall and a higher rate of h-affable negative samples degrades precision. Investigating these phenomena further, we find that an early paper on adversarial training in image classification (Kurakin et al., 2017) reports that adversarial training lowers the accuracy on the original clean test examples even though it improves on the adversarial samples (generated through automatic perturbation).

## 5.2 Data curation for relation extraction

As previously mentioned, relation extraction tasks that extract multiple entities from a single paragraph or abstract can have naturally occurring high rates of h-adversarials and h-affables. Hence, for relation extraction where multiple training samples can be generated from the same source text, we recommend that annotators curate at most 1 training sample for each type of relationship from a given source text. Drawing multiple training samples from the same source text can result in more h-affables or excessively increase the rate of h-adversarials, leading to degraded performance. In addition, explicitly annotating negative samples from text that is not part of the positive samples can potentially improve performance by lowering the h-adversarial rate and increasing the percentage of random samples.

## 5.3 Consistency of rule of thumb

Our experiments show that generally low to moderate (as a thumb rule 10 - 30%) h-adversarial rates improve generalization performance, although need not be always the case. For instance, Ta-

ble 5 shows that when RoBERTa is trained on IMDB-C-5H and IMDB-C-2K, peak performance on SemEval dataset is observed when there no h-adversarials, i.e, @$r_{hv}^{pn}$=0.0. This phenomena where introducing no h-adversarials can sometimes potentially achieve good performance shows the stochastic nature of neural network training procedure and that h-adversarials may not always be able to guide the neural network to learn the discriminatory features.

## 6 Related works

### 6.1 Data augmentation strategies to improve robustness

Data augmentation strategies involving minor perturbations to the input samples can be label altering or label preserving to improve robustness (Wang et al., 2022). Label altering perturbations can be either manually created (Kaushik et al., 2020), or generated using generative adversarial networks (Robeer et al., 2021). Label preserving strategies (Jin et al., 2020; Wei and Zou, 2019; Feng et al., 2021) are far more common as they are easier to automate, and may have some benefits particularly in low-resource settings such as specific domains (Wang et al., 2020). Label altering perturbations are known to improve robustness (Kaushik et al., 2020; Udomcharoenchaikit et al., 2022).

### 6.2 BERT Generalizability

McCoy et al., 2020 train 100 BERT models on natural language inference (NLI) task using MNLI dataset (Williams et al., 2018) with the exact same settings and dataset and find that the results are

fairly consistent; ranging from 83.6% to 84.8% when evaluated on MNLI test set. However, the performance varied substantially when evaluated on the distinct HANS dataset with accuracy ranging from 4% to 76% (McCoy et al., 2020). This highlights a key challenge in relying on test set performance as the primary indicator of generalizability, *i.e.*, the results may not be reproducible in a dataset other than the official test set. One explanation that McCoy et al. (2019) suggest is that BERT models learn correlations such as lexical overlap from the training data, and take advantage of the same correlations on the test data.

Elangovan et al. (2021) find that higher percentage overlap in the unigrams, bigrams and trigrams between train and test can also cause inflated test set performance. The adversarial GLUE benchmark (Wang et al., 2021) shows a 52 point drop on BERT models compared to evaluating against the GLUE benchmark (Wang et al., 2018). Thus, models learning shortcuts or spurious correlations will not be able to generalize to other real-world data, which, however, will not be reflected through the evaluation using relatively simple test sets. Strategies such as adding h-adversarial instances may support improved robustness.

## 7 Conclusion

Data annotation and curation is the most human intensive, expensive and time consuming aspect of developing machine learning applications. Curating large volumes of data without understanding how the data distribution contributes to improving a model can result in misdirected effort. In addition, the resulting models with poor generalization performance may be inadequate to automate or aid manual curation efforts. Hence, understanding the characteristics of training samples is key to ensuring that the right variety of samples are collected to create robust machine learning applications. In our work, we find that additional similar examples sharing a label, h-affables, may not contribute to improving performance despite increasing training data quantity. Similar examples with divergent labels, h-adversarials, add more value.

Our work contributes insights into the preparation of high-quality training data to support robust model generalization. Specifically, we demonstrate how moderate quantities of h-adversarials in combination with more diverse samples can substantially improve robustness.

## Limitations

Our work investigates BERT generalizability in the context of a small set of text classification and relation extraction tasks. Our findings empirically indicate how 10-30% of h-adversarial rate improves performance, especially when the baseline (h-adversarial rate 0%) performance is low. Further investigation is required to understand why increasing the h-adversarial rate further results in performance plateaus or even degradation. Further analysis is also required for other tasks such as Named Entity Recognition (NER) and Question Answering (QA) tasks as well as multi-class classification problems. Finally, implications for the increasingly high-parameter large language models remain to be explored.

## Ethics Statement

We comply with the ACL Ethics Policy[6]. This work has created a resource from public repositories (PubMed), or re-analyzed existing datasets. No human annotation was performed. Our focus is on empirical study of model robustness under generalization.

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

# A Appendix

## A.1 Algorithm Pseudo-code

---
**Algorithm 1** Keywords And-Or detection

---
1: **procedure** LABELABSTRACT(*abstract*)
2:     $set_a \leftarrow$ ["activation", "trigger", "interact", "inhibit", "regulate", "suppress"]
3:     $set_b \leftarrow$ ["gene", "protein", "chemical" ]
4:     $abstract \leftarrow$ lower($abstract$)
5:     $has\_set_a \leftarrow$ any($set_a$) in $abstract$
6:     $has\_pair\_set_b \leftarrow$ anyTwo($set_b$) in $abstract$
7:     **if** $has\_set_a$ and $has\_pair\_set_b$ **then**
8:         $label \leftarrow$ Positive
9:     **else**
10:        $label \leftarrow$ Negative
11:     **end if**
12:     **return** $label$
13: **end procedure**

---

## A.2 KDAO H-adversarial sample generation code

### A.2.1 Transform a positive sample into negative sample

```python
keywords1 = [''activation'', ''trigger'',
''interact'', ''inhibit'',
''regulate'', ''suppress'']

keywords2 = [''gene'', ''protein'', ''chemical'']

def randomly_substitute_keywords(x):
    words = [ w for w in x.split('' '') if not any(k.lower() in w.lower()
        for k in keywords1+keywords2)]

    key_i = np.random.choice([0,1])
    if key_i == 0:
        for k in keywords1:
            insensitive = re.compile(re.escape(k), re.IGNORECASE)
            w =  np.random.randint(0,len(words)-1)
            x = insensitive.sub(words[w], x)
    else:
        for k in keywords2:
            insensitive = re.compile(re.escape(k), re.IGNORECASE)
            w =  np.random.randint(0,len(words)-1)
            x = insensitive.sub(words[w], x)
    return x
```

### A.2.2 Transform a negative sample into positive sample

```python
keywords1 = [''activation'', ''trigger'',
''interact'', ''inhibit'',
''regulate'', ''suppress'']

keywords2 = [''gene'', ''protein'', ''chemical'']

def randomly_add_keywords(x):

    key_i1 = np.random.randint(0,len(keywords1)-1)
    key_i2 = np.random.randint(0,len(keywords2)-1)

    key_1 = keywords1[key_i1]
    keys_2 = keywords2[:key_i2] + keywords2[key_i2+1:]

    words = x.split('' '')
    l1 =  np.random.randint(0,len(words)-1)
    l2 =  np.random.randint(0,len(words)-1)
    l3 =  np.random.randint(0,len(words)-1)
```

```
    words.insert(l1, key_1)
    words.insert(l2, keys_2[0])
    words.insert(l3, keys_2[1])

    return '' ''.join(words)
```

## A.3 KDAO H-affables sample generation code

```
def randomly_add_words(x):

    words = x.split('' '')
    l1 =  np.random.randint(0,len(words)-1)
    l2 =  np.random.randint(0,len(words)-1)
    l3 =  np.random.randint(0,len(words)-1)

    w1 = np.random.randint(0,len(words)-1)
    words.insert(l1, words[w1])

    w2 = np.random.randint(0,len(words)-1)
    words.insert(l2, words[w2])

    w3 = np.random.randint(0,len(words)-1)
    words.insert(l3, words[w3])

    return '' ''.join(words)
```

## A.4 BERT training and inference

### A.4.1 Infrastructure and cost

BERT model was trained on Amazon p3.2xlarge instance for around 2 to 10 hours each depending on the training data set. In addition, inference on 500,000 record took 1 each for each run and was run on 5 g4dn.xlarge instances in parallel. The cost of p3.2xlarge is 3.06 USD an hour and g4dn.xlarge is 0.526 USD per hour.

## A.5 Software Packages used

We used the following packages

- matplotlib==3.4.3

- sagemaker==2.115.0

- scikit-plot==0.3.7

- pandas==1.1.4

- torch==1.4.0

- transformers==4.3.3

- scikit-learn==0.24.1

- scipy<=1.6.0