# OpenReview forum: "Effects of Human Adversarial and Affable Samples on BERT Generalization"
_EMNLP/2023/Conference — EMNLP 2023 Findings_

### Official Review · Reviewer_6RJa · 2023-08-02

**Typos Grammar Style And Presentation Improvements:** Equations in L225 - L227 are not alig…
**Soundness:** 2

**Excitement:**

2: Mediocre: This paper makes marginal contributions (vs non-contemporaneous work), so I would rather not see it in the conference.

**Paper Topic And Main Contributions:**

This paper proposes two characteristics of training samples, i.e., human adversarial and human affable ones. It mainly investigates the question of how these two types of samples affect the generalizability of the BERT model.

**Questions For The Authors:**

Please refer to the section of "Reasons To Reject".

**Reasons To Accept:**

1. The empirical study of this paper is comprehensive. This paper focuses on two NLP tasks including text classification and relation extraction. And it uses two representative BERT for corresponding tasks. The illustration results from Figure 1 to 4 are helpful for learning the results in analysis.

**Reasons To Reject:**

1. There is a lack of discussion on what motivates the design of h-adversarials and h-affables. For example, in traditional adversarial training, the motivation for designing adversarial examples is that there exist samples around an original sample that makes the model change its prediction easily so we should include them for training. But for this paper, although h-adversarials and h-affables are defined in the beginning of the paper, it is unclear what they mean actually in training and why they matter. This is an important question for the general audience to understand the main idea of the paper, but unfortunately, it is not answered in the paper. For instance, it is still hard to get the idea of H-adversarial even taking a careful look at Table 1. And the alien language looks really weird and not helpful for the general audience to catch the idea in the beginning of the paper.
2. The paper seems to pay more attention to presenting an empirical study of the effect of h-adversarials and h-affables for NLP model generalization. The discussion in Section 5 is limited given that there is a lack of novel methods for improving generalization ability based on the observation in Section 4. Actually, for a general audience, they will be benefitted more from reasonable treatment in future research and it is unlikely for them to reimplement the empirical study.
3. There is a lack of discussion on what 10-30% extra h-adversarial can cost in training. For example, do they lead to extra training cost that is unbearable in real-world implementation?  And how much time cost are needed for generation? This should be covered despite the benefit it may bring in training.
4. There is a lack of comparison with other data augmentation methods in experiments, e.g., label altering and label preserving.

**Reproducibility:**

3: Could reproduce the results with some difficulty. The settings of parameters are underspecified or subjectively determined; the training/evaluation data are not widely available.

**Reviewer Confidence:**

3: Pretty sure, but there's a chance I missed something. Although I have a good feel for this area in general, I did not carefully check the paper's details, e.g., the math, experimental design, or novelty.

---

> ### Author Rebuttal · Authors · 2023-08-29
>
> Thank you for reviewing our paper. Our paper is an analysis paper and the main contribution and novelty is the insights into the behavior of h-adversarials and h-affables at varying proportions.
>
> 1.	H-adversarials training samples are known by many other names in literature as “counterfactuals”, “label-altering perturbations”, “contrast-sets”, there is no single consistent terminology to represent such samples (mentioned in Line 76-79) and our main contribution is understanding how various proportions of such samples affect generalization performance. The motivation behind of h-adversarial is that subtle differences in the input that changes the target labels allows the model to understand the linguistic features instead of relying of spurious features that machine learning models often do. For instance, consider the example “Book is good” à Positive sentiment, “Book is awful” à Negative, then it is clear then the terms ‘Book is” is NOT a discriminatory feature that changes the sentiment, but “good” and “awful” are words that change the sentiment, allowing the model to quickly learn key linguistic features. We will explicitly make the terminology and the context of h-adversarials much clearer. H-affables are label-preserving perturbations, e.g., “The book is good”, “The book is great” both are positive examples of sentiment analysis task.  The motivation behind H-affables is that common data augmentation strategies in NLP, use paraphrasing, replacing words with synonyms so that the label remains the same. Hence, in our work we study their effectiveness in improving model generalizability.
> As you have suggested, if the terminology and description is creating confusion among readers, we will definitely improve them inanition to the existing examples in table 2 for relation extraction and sentiment analysis task. We can also emphasize that our key contribution is understanding how various proportions of such samples affect generalization performance.
>
> 2.	Our novelty and contribution *involves analyzing how various proportions of h-adversarials and h-affables affect generalizability* as mentioned in Line 90, and there has been little or no investigation  into how various ratios of these samples affect generalization.  Our observation, that is consistent across 5 datasets in 2 domains evaluated across 5 random seed runs on very large test sets (6000 to 500,000 samples for various datasets) indicates that low to moderate h-adversarials improve generalizability by 10-20 points and h-affables don’t. We also show that only low to moderate proportions in combination with diverse samples help in generalization. This will give guidance and help in future dataset curation process when there is limited resources (e.g., labour, monetary) to annotate limited samples.
>
> 3.	10-30% extra h-adversarial samples does not add computation cost (GPU machine cost) since in the experiments,  the total number of training sample is kept the same -- we simply replace a subset of randomly selected random diverse samples with h-adversarials. We aim to analyze when we have fixed number of training samples, how the proportion of h-adversarials/h-affables affect the results. In terms of human annotation cost, it depends on whether h-adversarials were automatically or manually generated. Manual generation of h-adversarials would be expensive, as we have to rely on human expertise to  construct such samples as opposed to simply labelling them, hence all the more reason to be optimal with the proportion of such samples. Kaushik et al paper (cited in our works) that produced the IMDB-C dataset uses 90% h-adversarial rate, which can be optimized further, as our results demonstrate that peak performance can be achieved just 10% h-adversarials according to our results.
>
> 4.	Thank you for your suggestion, we can include report existing work. Existing data augmentation approaches such as “An Empirical Survey of Data Augmentation for Limited Data Learning in NLP” report at most 2-3 points boost on news classification datasets using label preserving methods. One challenge we have is that we can only report experimental performance on datasets that have label-altering (h-adversarial`) training samples with corresponding cross domain test sets to evaluate generalization performance for an apples to apples comparison. The only one we found to the best of our knowledge is Kaushik et al ICLR 2020 paper (cited in our works), but we could not report their performance as we are could not get them to confirm the version of Amazon, SemEval and Yelp test sets they were using. Secondly, Kaushik et al 2020 report single run results, hence we are not sure whether they reported peak / average / random performance across one or several runs. We can include their results with an astrix as we are not sure of which version of test-sets they have used. Regardless, our analysis is more about how various proportions of such sample impact performance while Kaushik’s work assumes over 90% h-adversarial rate.
>
>
> 4. Thank for pointing this out. We will fix the alignments.

---

### Official Review · Reviewer_dy3Y · 2023-08-04

**Typos Grammar Style And Presentation Improvements:** 1.  The textual information with high…
**Soundness:** 2

**Excitement:**

2: Mediocre: This paper makes marginal contributions (vs non-contemporaneous work), so I would rather not see it in the conference.

**Missing References:**

[1] Altinisik, Enes, et al. "Impact of adversarial training on robustness and generalizability of language models." arXiv preprint arXiv:2211.05523 (2022).

**Paper Topic And Main Contributions:**

In this paper, since improving the generalization of BERT in real-world scenarios, the authors consider two data augmentation methods including generating samples with seemingly minor differences but different ground-truth labels (h-adversarial) and the samples are quite minor differences meanwhile the same ground-truth labels (h-affable). The authors conduct experiments on text classification, relation extraction and keyword detection to discuss the impacts of model performance and generalization via augmenting h-adversarial and h-affable instances.

**Reasons To Accept:**

1. This paper clearly claims their contribution and limitation.
2. The authors make a discussion in detail after analyzing different h-adversarials/affables rates in a fixed training size and the impact of varying training sizes.
3. The experimental results provide beneficial insights into data annotation and curation procedures when employing BERT in a real-world context.


**Reasons To Reject:**

1. The technical novelty of generating adversarial and affable samples is limited. And only one approach to discussing the effects of human adversarial and affable samples seems not enough representative.
2. The motivation is ambiguous, the authors mentioned that "limited training data is acknowledged as a key obstacle to achieving generalizability". However, they examine the impact of data quality, the reason the considering data quality instead of quantity needs to be claimed.
3. The description of generating h-adversarials and h-affables is unclear.
4. The experiment lacks a comparison between the proposed approach and existing data augmentation strategies.


**Reproducibility:**

3: Could reproduce the results with some difficulty. The settings of parameters are underspecified or subjectively determined; the training/evaluation data are not widely available.

**Reviewer Confidence:**

2: Willing to defend my evaluation, but it is fairly likely that I missed some details, didn't understand some central points, or can't be sure about the novelty of the work.

---

> ### Author Rebuttal · Authors · 2023-08-29
>
> Thank you for reviewing our paper. Our paper is an analysis paper and the main contribution and novelty is the insights into the behavior of h-adversarials and h-affables at varying proportions.
>
> 1.	Our main contribution and novelty is to examine/ analyse the impact of various proportions of h-adversarials and h-affables on generalizability, whereas proposing methods to generate such h-adversarial of h-affable samples, either manually or automatically, is covered in other papers [Kaushik et-al 2020, Wang et al 2022, both cited on our paper] and in papers such as “Generating Realistic Natural Language Counterfactuals (Robeer et al., Findings 2021)” which we can also include and explicitly call out that generation of such samples is covered in previous works.
>
> 2.	Our motivation, is to improve robustness (generalization) when we have limited resources for data curation. In such cases, only limited quantity of data can be curated, then the only other lever is to improve the data quality to mitigate the effects of limited data. More specifically, our contribution is analyzing the impact of how various proportions of h-adversarials and h-affables affect generalization performance. For instance, does robustness improve as we add more h-adversarials, is it worth spending more time collecting and annotating such samples? Our experiments indicate that this is not the case and only low to moderate proportions in combination with diverse samples help in generalization. The significance of our contribution is in analyzing the impact of such samples, allowing curators to focus on annotating the right samples given a fixed number of samples to annotate (a real-world constraint as cost is proportional to how many samples are required, especially in expert domains) to improve generalizability. Hence, in our experiments, in-order to evaluate generalizability we have used cross domain datasets or test datasets not sourced from the same single split where possible.  In our datasets we have ensured this for 4/5 datasets, except for ChemProt dataset where we could not find an alternative test set. Our experiments have been conducted quite rigorously on very large test sets ranging from 6000 to 500,000 samples with 5 random seed runs across 5 datasets in 2 different domains across 2 different tasks, with reported standard error and the results have been consistent.
>
>
>
> 3.	Thank you for providing feedback on how to improve the readability of our paper. We will add clearer descriptions of how h-adversarials and h-affables are generated, in addition to existing examples on table 2, subsections “Automatic generation of h-adversarials & h- affables.”  under “3.1.2 Self-Labelled Keywords And-Or detection” and “3.1.3 Chemprot”.   We can also cover existing works in “Related works” section and provide a systematic summary of existing methods used to generate such h-adversarial or h-affable samples, citing works of (Kaushik et-al ICLR 2020, Wang et al 2022, both cited on our paper] and as well as “Generating Realistic Natural Language Counterfactuals (Robeer et al., Findings 2021)”.
>
> 4. Thanks for your suggestion. We can expand  on  data augmentation techniques is related to our paper. Work such as “An Empirical Survey of Data Augmentation for Limited Data Learning in NLP” Chen et al 2023) ” report at most 2-3 points boost on news classification datasets using label preserving methods. We can explicitly cover this in our related works section, in addition to what is already covered in “section 6 - Data augmentation strategies to improve robustness.”  Our paper is an analysis paper that investigates how various proportion of h-adversarials and h-affables affect generalizability. We report experimental performance on  where the training datasets that have label-altering (h-adversarial`) samples with a corresponding cross domain test-set to evaluate generalization.
>
> 4.	Thank you pointing out the need to highlight content in Table 1. We will include that

---

### Official Review · Reviewer_V7ws · 2023-08-05

**Soundness:** 4

**Excitement:**

4: Strong: This paper deepens the understanding of some phenomenon or lowers the barriers to an existing research direction.

**Paper Topic And Main Contributions:**

The paper aims to understand the effects of similar examples and their labels in an adversarial training set for better generalization across 3 tasks, for BERT models.

**Reasons To Accept:**

* The paper does an exhaustive analysis of the types of examples, and finds that with a moderate number of similar examples with differing labels, generalization is better
* With more similar examples that have the same labels, spurious correlations degrade generalization


**Reasons To Reject:**

* The analysis can be improved by comparing the h-adversarial and h-affable ratios in the generalization datasets through auto evaluation metrics. The rule of thumb of 10-30% might vary based on the generalization datasets we evaluate on.

**Reproducibility:**

4: Could mostly reproduce the results, but there may be some variation because of sample variance or minor variations in their interpretation of the protocol or method.

**Reviewer Confidence:**

4: Quite sure. I tried to check the important points carefully. It's unlikely, though conceivable, that I missed something that should affect my ratings.

---

> ### Author Rebuttal · Authors · 2023-08-29
>
> Thank you so much for your review comments.
>
> We agree with your review comments that the rule of thumb of 10-30% might vary based on the generalization datasets used and we will explicitly call that out in our discussion in our manuscript. Our key take away is that low to moderate proportions of h-adversarials help improve generalization in combination with diverse random samples and the exact proportion will vary depending on the dataset as you have rightly pointed out.

---

### Meta-Review · Area_Chair_WBU9 · 2023-09-22

**Recommendation:** 3

**Metareview:**

This paper studies the effect of two types of data manipulation on the generalization ability of Bert models. Reviewers gave different recommendations for this paper. R1 found this paper insightful to study the interaction between these two augmentations. R2 and R3 found this paper rather narrow due to concerns about the training size, unclear comparison, and limited tasks. The AC read the full paper and the discussions and stood with R1. The motivation and the scope of this paper have been clearly described, and the authors conducted most of their study under a fixed training size. Increasing the training size leads to even more improvements. Hence, there AC deemed this paper made good empirical discovery and should be accepted to Findings.

---

### Decision · Program_Chairs · 2023-10-07

**Decision:**

Accept-Findings

**Comment:**

This paper studies the effect of two types of data manipulation on the generalization ability of Bert models. Reviewers gave different recommendations for this paper. R1 found this paper insightful to study the interaction between these two augmentations. R2 and R3 found this paper rather narrow due to concerns about the training size, unclear comparison, and limited tasks. The AC read the full paper and the discussions and stood with R1. The motivation and the scope of this paper have been clearly described, and the authors conducted most of their study under a fixed training size. Increasing the training size leads to even more improvements. Hence, there AC deemed this paper made good empirical discovery and should be accepted to Findings.